# mRNA Vaccine Technology Beyond COVID-19

**DOI:** 10.3390/vaccines13060601

**Published:** 2025-05-31

**Authors:** Sola Oloruntimehin, Florence Akinyi, Michael Paul, Olumuyiwa Ariyo

**Affiliations:** 1Molecular Virology Laboratory, First Moscow State Medical University (Sechenov), 119991 Moscow, Russia; 2SmartSciFrika, Mainland Garden Estate, Shagamu 121101, Ogun State, Nigeria; florencetaduda@gmail.com (F.A.); eshioramhe@gmail.com (M.P.); olumuyiwaariyo@gmail.com (O.A.); 3Kenya Medical Research Institute—Wellcome Trust, Kilifi P.O BOX 230, Kenya; 4The West African Center for Emerging Infectious Diseases, Jos University Teaching Hospital, Jos 930241, Plateau State, Nigeria; 5Department of Medicine, Afe Babalola University, Ado-Ekiti 360102, Ekiti State, Nigeria

**Keywords:** mRNA vaccine, infectious diseases, cancer, personalized medicine, rare and genetic diseases, dendritic cells, lipid nanoparticles, exosomes

## Abstract

Background/Objectives: Since their approval in early 2020, mRNA vaccines have gained significant attention since the COVID-19 pandemic as a potential therapeutic approach to tackle several infectious diseases. This article aims to review the current state of mRNA vaccine technology and its use against other diseases. Methods: To obtain accurate and reliable data, we carefully searched the clinicaltrial.gov and individual companies’ websites for current ongoing clinical trials reports. Also, we accessed different NCBI databases for recent articles or reports of clinical trials, innovative design of mRNA vaccines, and reviews. Results: Significant progress has been made in the design and improvement of mRNA vaccine technology. Currently, there are hundreds of ongoing clinical trials on mRNA vaccines against different cancer types, infectious diseases, and genetic and rare diseases, which showcase the advancement in this technology and their potential therapeutic advantages over traditional vaccine platforms. Finally, we predict what could be a potential future direction in designing more effective mRNA vaccines, particularly against cancer. Conclusions: The results of many of the ongoing clinical trials have shown significant positive outcomes, with many of the trials already at Phase III. Despite this outlook, however, some have been terminated or withdrawn for several reasons, some of which are not made available. This means that despite the advancement, there is a need for more research and critical evaluation of each innovation to better understand their immunological benefits and long-term effects.

## 1. Introduction

The first mRNA therapeutic attempt was designed in 1995 to target cancer, loading the protein on dendritic cells (DCs) [1,2]. After this breakthrough research by Hsu and his colleagues, the technology of mRNA vaccines seemed to be in the shadow until the emergence of the COVID-19 pandemic, when mRNA vaccines were rolled out to tackle the disease, which became the fastest vaccines to be approved during any pandemic in history. Despite this great success, the liquid nanoparticle (LNP)-based mRNA vaccines were not without their drawbacks. Besides the need for storage and transportation under very low temperatures, the associated cytokine storm that leads to many adverse immunologic effects has been cause for serious concern [3,4]. However, given their potential, many companies and institutions have been working to advance this technology and target other infectious and genetic diseases.

Some of the strategies to improve mRNA vaccines, besides untranslated regions (UTR), codon optimisations, and nucleotide modifications, which were previously employed by many companies including Moderna and Pfizer in their vaccine designs, include more recent approaches such as alternative delivery systems, including polymer-based delivery systems, exosome/peptide, ionisable lipids [5], and the use of next-generation lipid nanoparticles. Other advancements are the use of self-amplifying mRNA (sa-mRNA), thermostability, adjuvant integration, the design of personalized and targeted vaccines, and the use of dendritic cell-based mRNA vaccines, which seems to have gained more attention among other strategies [6,7,8,9].

Currently, numerous clinical trials are ongoing with the use of these innovative mRNA vaccines as therapies for infectious diseases such as influenza, rabies, Zika virus, Epstein–Barr Virus (EBV), Human Metapneumovirus (hMPV), Respiratory Syncytial Virus (RSV), Herpes Simplex Virus (HSV), and HIV. Available clinical trial data indicate that a significant number of genetic and rare diseases are being investigated for potential cures using mRNA therapeutics, and various cancer types have also been targeted with mRNA vaccine formulations (see Table 2 and Appendix A).

## 2. The Structure and Mechanism of Action of mRNA Vaccines

mRNA vaccines contain synthetic mRNA molecules that direct the production of antigens to elicit immune responses. Structurally, the In Vitro transcribed mRNA mimics the endogenous mRNA. The two main mRNA types under investigation as vaccine antigens are self-amplifying RNA and non-replicating mRNA [10]. The non-replicating (conventional) mRNA contains five components: 5′ cap, 5′ untranslated regions (UTR), an open reading frame (ORF), 3′ UTR, and 3′ end poly(A) tail. The self-amplifying mRNA contains all these components with an extra coding region in their ORF that codes for the viral replication machinery, allowing for continuous intracellular RNA amplification and subsequent increased antigen production [11]. The 5ʹcap contains 7-methylguanosine nucleoside (m7G) linked to the 5′-triphosphate bridge. The cap increases the stability of mRNA against 5′ exonucleolytic degradation. Next to the 5′ cap, the 5′ UTR contains important regulatory regions, such as the internal ribosome entry site (IRES) and ribosome binding site (RBS). These components recruit the ribosome and other translation factors to initiate and regulate translation. The open reading frame is the most important part of the vaccine as it codes for the area that is translated into the protein. The overall translation efficiency of mRNA can be maximized by selecting the right codons in this region. This can be conducted by replacing rarely used codons with more frequently occurring codons that encode the same amino acid residue [12,13]. The polyadenylated tail at the 3′ end regulates the stability and translation efficiency of the mRNA by interacting with proteins that control its degradation and translation initiation (see Figure 1). Modified nucleosides are added to enhance translation. This modification prevents recognition by the antigen-presenting cells, leading to a high level of translation and producing a significant number of needed proteins/antigens [14]. The mRNA is encapsulated with lipid nanoparticles for efficient delivery into the cells.

Upon injection of the vaccine, the muscular cells take up the LNP-mRNA. It escapes the endosome, releasing the mRNA into the cytoplasm [15]. Following its release, the host ribosome recognizes it and translates it into protein/antigen. The protein can be cleaved by the proteasome into peptides or transported outside the cell by the Golgi apparatus [16]. The danger-associated signals produced by the LNP recruit antigen-presenting cells like dendritic cells and neutrophils, which present the peptides to cytotoxic T lymphocytes on the cell surface by the major histocompatibility complex (MHC) class I. MHC class II proteins can also present released antigens on the cell surface to helper T lymphocytes after being absorbed by cells and broken down inside endosomes [17,18]. As a result, CD8T cells generate humoral immunity that can neutralize subsequent infections from the same pathogen and establish a rapid immune response. Antigen-specific antibodies are produced by B cells and are triggered by either direct antigen recognition at the B cell receptor (BCR) or CD4T cell assistance [13,19]. Later, these antibodies can identify the same antigen in subsequent exposures, quickly triggering an immune response that prevents infection and growth. It is this humoral defence that brings long-term vaccine-mediated immunity.

### Key Advantages over Traditional Vaccine Platforms

Unlike conventional vaccines (inactivated, live-attenuated, subunit, recombinant, and other conjugate types), mRNA vaccines offer several distinct advantages. In the pioneer study by Kariko and his colleagues, modified nucleosides m5C, m6A, m5U, s2U, or pseudouridine was first tested [20]. First, the mRNA vaccine does not contain the pathogen itself. Hence, there is no chance of infection after vaccination. The cell-free in vitro production also reduces safety issues such as viral contamination and cell-derived impurities commonly present in other platforms [11]. Second, since mRNA expression does not require nuclear entrance, it is safer than DNA-based vaccines and has a nearly minimal chance of random genome integration [21]. Third, its ability to encode several distinct antigens in a single formulation enables accurate and strong immune responses. It promotes humoral and cellular immune responses and induces the innate immune system. Fourth, the mRNA vaccine platform’s adaptability is particularly beneficial for manufacturing since it allows for standardization of production, as changes to the encoded antigen do not affect the physical–chemical properties of the mRNA backbone [22]. Fifth, it has a wide range of applications. mRNA technology treats infectious diseases, genetic diseases, cancer, and diabetes. Lastly, mRNA is transiently active; hence, it is readily degraded by the metabolic pathways [23].

## 3. Lipid Nanoparticle (LNP) Delivery Systems

### 3.1. Advances in Lipid Nanoparticle (LNP) Delivery Systems

Lipid nanoparticles (LNPs) are non-viral vectors that have gained wide application as efficient delivery tools. They exhibit low cytotoxicity and immunogenicity, high tissue penetration, high nucleic acid encapsulation, and efficient transfection with good biocompatibility [24]. Structurally, LNP consists of phospholipids, cholesterol, and polyethylene glycol (PEG) containing lipids. Phospholipids improve stability, enhance the ease of mRNA encapsulation, and promote cellular absorption. Ionizable lipids possess unique pH-responsive properties that enable them to gain a positive charge in acidic environments, such as endosomes, while remaining neutral during systemic circulation [25]. They are neutral at physiological pH, encouraging stable LNP production, prolonging bloodstream circulation, and inhibiting immune cell clearance. In the acidic environment, they become protonated (positively charged) and directly interact with the negatively charged phospholipids, promoting the release of cargo into the cytoplasm and aiding in the rupture of the endosomal membrane. Cholesterol stabilizes LNP structure, regulates the fluidity and permeability of membranes, and improves particle stability [26,27].

### 3.2. Types of Lipid Nanoparticles (LNPs)

Lipid nanoparticles (LNPs) are classified into four groups: liposomes, solid lipid nanoparticles (SLNs), nanostructured lipid carriers (NLCs), and hybrid lipid–polymeric nanoparticles. Liposomes serve as the foundational version of lipid nanoparticle (LNP) technology. They are made up of a phospholipid bilayer, primarily composed of phosphatidylcholine (PC) and other phospholipids. This structure self-assembles into bilayers because of the amphiphilic properties of the lipids, which create an aqueous core surrounded by hydrocarbon chains. This unique configuration highlights the potential of liposomes in various applications, paving the way for advanced drug delivery systems [2,28]. The aqueous interior of liposomes contains hydrophilic drugs, as the hydrocarbon chain portion of the lipid bilayer entraps hydrophobic drugs. Its size ranges between 20 and 1000 nm and can be synthesized as unilamellar or multilamellar. Small Unilamellar Vesicles (SUVs) (20–100 nm) consist of a single bilayer, while large Unilamellar Vesicles (LUVs) (100–500 nm) have a single bilayer but a larger encapsulation volume. Multilamellar Vesicles (MLVs) (0.1–10 µm) composed of multiple concentric bilayers. The size determines the half-life and drug encapsulation of the liposome. The small unilamellar liposomes exhibit higher encapsulation efficiency, increased drug half-life, and the ability to evade the immune system [29]. Liposomes are difficult to produce at a large scale due to the complex production process that involves organic solvents. To address these challenges, solid and nanostructured lipid nanoparticles have been developed.

Solid lipid nanoparticles (SLNs) are solid colloidal particles that range in size from 4 to 1000 nm. They contain physiological, biodegradable, and biocompatible lipids and surfactants in their formulation, which can incorporate both lipophilic and hydrophilic drugs inside the lipid matrix [26,30]. The solid lipid serves as the dispersed phase, while the surfactant acts as an emulsifier. The solid lipid is typically made from triglycerides, glyceride mixtures, or even waxes, and it stays solid at room and body temperature; the surfactant concentration range is usually 0.5 to 5% (*w*/*v*) to improve formulation stability [2,31]. The selection of lipids and surfactants significantly impacts the nanoparticle formulation’s physicochemical properties. The structure of solid lipid nanoparticles (SLNs) is influenced and determined by the production method and the solubility of the components and component formulation. Three structural types have been described: the homogenous matrix model, the drug-enriched shell, and the drug-enriched core. A homogenous matrix model is used for highly lipophilic drugs; the particles are created using a cold or hot homogenization process. The drug is either dissolved in a lipid matrix, a high-pressure homogenization process, and mechanical breakings leading to nanoparticle formation, or the lipid is dissolved in a lipid matrix with increased temperatures, leading to nanoparticle formation. It offers slow and controlled drug release. The drug-enriched shell is produced by hot homogenization as the system cools down the lipid precipitate, forming the lipid core. The concentration of the drug increases until its solubility limit is reached. When this limit is reached, the mix of drug and melted lipid crystallizes to form an outer shell. It results in a faster release of drugs. In the drug-enriched core, the drug crystallizes before the lipid and becomes trapped in the core, with a pure lipid shell forming around it; this occurs when the drug has higher solubility in the molten lipid but separates during solidification [27,31]. Unlike liposomes, SLNs have prolonged release and better stability. However, the solid lipid’s crystalline structure causes its poor integration rate, and it tends to gel. NLCs are developed to overcome the disadvantages of SLNs.

Nanostructured lipid carriers (NLCs) are modified forms of SLNs that contain a mixture of solid and liquid phases (oil), forming a formless matrix, improving stability and capacity loading [24,29,30]. The lipids used here are biologically compatible, hence reducing toxicity. Depending on their structure, they can be classified into three types: The imperfect crystal type comprises lipids with different chain lengths forming a matrix with several voids and imperfections that provide a more suitable environment for drug incorporation. The amorphous type is obtained using medium-chain-length triglycerides and solid lipids. The solid lipids do not undergo recrystallization, thus resulting in solid particles with an amorphous structure. The drug’s unwanted release is reduced, extending its shelf life. The multiple LNC has nanocompartments of oil containing dissolved drugs [29,32,33]. Methods like high-pressure homogenization, ultrasonication, spraying, dying, solvent injection, double emulsion, and microfluidics are used to prepare lipid nanoparticles. Microfluidics is a recent form that leads to the production of nanoparticles with uniform particles; it is a promising approach for the large-scale production of SLNs and NLCs [26,27].

### 3.3. Improvement in Delivery Systems

The FDA-approved COVID-19 vaccines (Pfizer, USA -BioNTech, Germany and Moderna, USA) utilized LNPs with proprietary lipids like ALC-0315 (Pfizer) and SM-102 (Moderna) [25]. These LNPs used ionizable lipids that self-assemble with mRNA at acidic pH, which promotes endosomal escape. Besides LNP, platforms such as improved ionized lipids, polymer-based delivery systems, peptide-based delivery, and exosome and extracellular vesicle (EV) delivery are being investigated to bypass LNP’s limitations. A 2022 study conducted by Kamerker and her colleagues demonstrated engineered exosomes delivered mRNA to pancreatic cancer cells with higher efficiency [34].

Cationic polymers, such as polyethyleneimine (PEI) and poly(amidoamine) (PAMAM), have numerous amino acids that provide a high density of positive charges for mRNA complexation and endosomal escape, hence a high delivery efficiency. It acts as a protonation buffer, allowing protons, chlorides, and water to enter the nucleus [12,35]. This causes osmotic expansion and more endosomal membrane disruption. Despite the delivery efficiency, they have high cytotoxicity levels. Non-cationic polymers like polyethylene glycol (PEG) and cyclodextrin (CD), which are biocompatible compounds, are conjugated to the cationic polymers to lower their charge density while maintaining their ability to condense nucleic acids, thus lowering the toxicity level. Considering the low biodegradability and biocompatibility of the current polymers, other polymeric carriers such as polyesters and dendrimers are being developed to overcome this limitation [35,36].

While mRNA vaccine for COVID-19 remains one of the greatest achievements in modern medicine, the idea of using this concept for cancer started over 20 years ago when Hsu and his colleagues demonstrated for the first time, the potential of using autologous tumour-specific antigen loaded on dendritic cells to induce antitumor immune response in cancer patients [1]. This groundbreaking clinical trial of using dendritic cells (DCs)-based vaccines in cancer laid a crucial foundation for what has now become a golden approach, and many researchers now focus on the use of monocyte-derived DCs (moDCs) generated in vitro for mRNA vaccine production. The efficacy of this type of DC has led to the approval of Sipuleucel-T (a DC-based vaccine) for prostate cancer by the FDA. Recently, however, due to extensive culture periods and supplements required to produce mature moDCs, many researchers have sought alternative ways of generating DCs. Myeloid DC (mDC) and plasmacytoid DC (pDC) are two alternatives that have shown promising immunologic and clinical outcomes in metastatic melanoma patients [7,9] and lung cancer [6]. pDCs have also been shown as promising tools for multispecific viral and tumour antigen-specific T cells [8]. For a comprehensive review on the use of dendritic-based vaccines in immunotherapy, we refer readers to a recent review on this subject [37]. Furthermore, to enhance targeted delivery, LNPs are now being conjugated with antibodies or peptides to enable cell-specific delivery. For example, targeting dendritic cells via CD40 ligands enhances antigen presentation, as shown in cancer vaccine trials [38]. Pulsing DCs with human cytomegalovirus (CMV) pp65-LAMP (lysosomal-associated membrane protein) mRNA to target gliomas is currently being investigated in the clinical trial (NCT03688178), and it will be interesting to know the outcome of the trial. This cytomegalovirus unique protein has been reported in a significant number of people suffering from malignant gliomas (MGs), and targeting the proteins within the cancer microenvironment provides alternative therapeutic potential to gliomas, and this approach might be relevant to other cancer types.

### 3.4. Strategies for Enhancing Stability and Immunogenicity

One challenge associated with lipid nanoparticles (LNPs) is the requirement for extremely low temperatures for storage and transportation, highlighting the need for improved storage conditions.

The ARCT-154 (Kostaive) vaccine, developed by CSL and Arcturus Therapeutics and recently approved in Japan, features an active ingredient called zapomeran, which is a self-amplifying messenger RNA (mRNA). This self-amplifying mRNA (also known as sa-mRNA) utilizes a mechanism that instructs cells to produce an enzyme replicase, enabling the generation of additional copies of mRNA upon delivery. In contrast to other approved COVID-19 vaccines, Kostaive can be stored at a higher temperature of approximately 2 °C. Additionally, the development of this sa-mRNA vaccine is reported to significantly reduce immunogenicity, as it requires smaller doses to elicit strong cellular immune responses. This approach effectively minimizes immune toxicity and other adverse effects commonly associated with other COVID-19 vaccines. Furthermore, mRNA-1283, a next-generation COVID vaccine developed by Moderna, is currently in Phase 3 trials and can be stored at temperatures ranging from about 2 °C to 5 °C [39]. Also, the addition of adjuvants to the mRNA vaccine is another notable innovation that would enhance the vaccine’s protection. Co-delivering immune stimulants (for example, TLR agonists) enhances responses. BioNTech’s FixVac platform incorporates uridine-rich mRNA to activate TLR7/8, boosting T-cell immunity in melanoma trials [40].

## 4. Applications Beyond COVID-19

### 4.1. Infectious Diseases

Infectious diseases remain a significant global threat to human health, particularly in low-income countries where emerging pathogens exist without approved vaccines or widely available treatments, resulting in devastating consequences [41,42], driving the call for extensive research in vaccine development [43,44]. Given that vaccines are a powerful means of enhancing global health security, several have been created to combat the threat of infectious diseases, significantly transforming disease prevention strategies [44,45,46]. With reduced concerns regarding insertional mutagenesis [11,47], remarkable progress has been made in applying mRNA technology for both infectious and non-infectious diseases. This technology has proven effective in treating cancer and infectious diseases [12]. The therapeutic and preventive roles of mRNA technology have been explored to support the treatment of infectious diseases. Recent advancements in mRNA design, delivery systems, and production scalability have expanded its applications to oncology, autoimmune disorders, and genetic diseases. The COVID-19 pandemic accelerated research into mRNA vaccines for other infectious diseases, highlighting their potential as a groundbreaking platform for vaccine development. This progress is particularly evident in the enhancement of delivery systems for these vaccines [42,45,48,49]. mRNA vaccines hold promising applications for various human infectious diseases caused by pathogens such as filoviruses (including Ebola), arboviruses (like Zika, Dengue, and Chikungunya), influenza, HIV, and rabies. Recent studies have underscored the potential of mRNA vaccines against the Zika virus (ZIKV). Starting with the first-generation mRNA-1325 Zika vaccine developed by Moderna, which is based on structural proteins from a 2007 Micronesia Zika virus isolate, to the triple repeat domain III mRNA vaccine, 3xEIII, encapsulated in lipid nanoparticles (LNPs), both elicited neutralizing antibodies and long-term immunity in mice [23]. Similarly, other Zika virus vaccines, including the mRNA vaccine encoding ZIKV prM-E proteins in LNPs, provided protection in AG129 mice against lethal ZIKV challenges [50]. A self-replicating mRNA vaccine encoding ZIKV prM-E proteins demonstrated efficacy in IFNAR1-/-C57BL/6 mice. However, its effectiveness was inversely related to the dose in wild-type mice due to type I interferon responses [51]. An optimized mRNA vaccine, mRNA-1893, produced comparable neutralizing antibody titers to its predecessor at a lower dose, ensuring complete protection in non-human primates [52]. The HIV vaccine development has faced extreme difficulties consequent to the challenge of the high variability of the viral envelope antigens of HIV, driving the huge genetic diversity of HIV, and the frequent escape mutations consistent with the virus [53,54]. The fact that most people infected with HIV are infected with about 20% of HIV variants further compounds the problem of successfully developing a vaccine with broad neutralizing antibodies (bNAb) [55,56,57]. Presently, the HIV-1 trimeric envelope (Env) glycoprotein (gp160/gp140) has been targeted by mRNA-1574 Moderna as a vaccine candidate, while the engineered gp120 (outer domain of Env) and stabilized HIV-1 Env gp140 trimer developed by the eOD-GT8 60mer mRNA (IAVI/Moderna) and BG505 MD39.3 mRNA (NIH/NIAID), respectively [[58],[59].[60]]. Currently, the HIV mRNA vaccines are being tested in healthy individuals and patients with HIV with the aim of passive immunization.

The human avian influenza (H5N1) virus has been reported in more than 23 countries since its first report and has remained a significant threat to global health, resulting in continuous seasonal economic loss. Global yearly seasonal influenza epidemics record about 3 million people developing illnesses, of which up to 600,000 died (mainly children) from influenza cases worldwide [61,62]. For a long time, embryonated chicken eggs have been the platform for the generation of influenza vaccines, although this approach presented greater scalability, effectiveness was about 60% due to the constantly changing antigenic drift and shift, limited duration of immunity, and antigenic mismatch between the selected vaccine strains and circulating influenza strains [63] since influenza mutation is seasonally influenced [64], posing a challenge for the development of a global influenza vaccine. Although the mRNA influenza vaccine targeting Hemagglutinin (HA) has long been reported [65], the most significant mRNA vaccines for influenza came after the success of the mRNA vaccine for the COVID-19 pandemic. The trivalent mRNA vaccine candidate with cross-specific humoral immune response in mice was reported by Elena et al. [66]. Others in clinical trials include the ARCT-2304 encoding viral glycoproteins (HA and neuraminidase) and the mRNA-1018 mRNA vaccine developed by Arcturus Therapeutics and Moderna, respectively [67]. The recent outbreak of influenza in the US has prompted the US Department of Health and Human Services (HHS) to announce GBP 590 million in support of Moderna to expedite the development of mRNA vaccines against potential pandemic flu viruses [67,68]. Several other companies and institutions are also developing mRNA vaccines for influenza and other infectious diseases (see Table 1).

### 4.2. Cancer Immunology

A recent publication by Chen et al. estimated the global economic cost of cancers to be at USD 25.2 trillion from 2020 to 2050, imposing a marked toll on the economy through reduced productivity, unemployment, labour losses, and capital investment reductions. Cancer treatments, including surgery, chemotherapy, radiation, and targeted therapies, have significantly improved patient outcomes in recent decades, but all these methods have come with limited outcomes [29,69], requiring a search for the alternative best option. Cancer immunotherapy’s desired outcome is to achieve long-lasting positive responses in patients; therefore, the quest to develop effective vaccines and therapy for cancer treatment has advanced lately with the introduction of several technologies and approaches in therapeutics [29,69,70]. Cancer vaccines target treatment rather than prevention, unlike vaccines for infectious diseases. Their goal is to stimulate the patient’s immune system using tumour antigens, triggering an antitumor response to help eliminate the tumour [71]. The successes of mRNA technology during COVID-19 revolutionized vaccinology and paved the way for numerous breakthroughs in disease treatment, particularly in cancer. The uniqueness of mRNA technology for cancer therapy includes the versatility and flexibility of mRNA to encode multiple antigens against diverse tumour-associated antigens (TAAs), the ability to induce both humoral and cellular immune responses, leading to broader and more robust protection against cancer in terms of vaccine, the use of mRNA to encode tumour-specific antigens (TSAs) allows the immune system to recognize and eliminate cancer cells displaying these antigens [29]. mRNA therapy offers numerous advantages over traditional cancer immunotherapy, primarily its ability to be used to achieve both monotherapy and synergistic effects for cancer immunotherapy, addressing the known features of tumour immune evasion and resistance, limited response rate and specificity, and immune-related adverse effects [70,72]. Issues of safety, delivery methods, and personalized treatments are addressed in most mRNA candidates for cancer immunotherapy [69]. Despite its advantage in inducing cross-presentation of multiple epitopes in antigen-presenting cells (APCs) with patient-specific human leukocyte antigens (HLAs), reducing HLA type restrictions and boosting T cell responses, a lot of research is still required in personalized mRNA cancer vaccines. Numerous mRNA cancer vaccines are currently in clinical trials, ranging from ex vivo mRNA-based DC vaccines to in vivo mRNA cancer vaccines. Promising results from clinical trials, such as the Lipo-MERIT trial, have demonstrated the effects of mRNA vaccines targeting melanoma-associated antigens, resulting in long-lasting immune responses in patients with advanced melanoma [40]. The Phase 3 clinical study of Adjuvant V940 (mRNA-4157) with Pembrolizumab shows promising results as a personalized cancer vaccine targeting tumour neoantigens in high-risk stage II-IV melanoma patients. This treatment significantly prolongs recurrence-free survival (RFS) and reduces the risk of recurrence or death by 44% [69,73].

Prostate cancer is the leading cancer among men in sub-Saharan Africa, with increasing incidence rates and high mortality rates [74]. Recent immunotherapies target either key tumour-associated antigens (TAA) or tumour-specific antigens (TSA). Tumour-associated antigens (TAA) are the most common prostate cancer antigens. They are characterized by weak tumour immunogenicity and specificity. To address this issue, mixtures of TAAs are being utilized to enhance the efficacy of cancer vaccines [70]. TAA serves as the target for the multivalent mRNA vaccine, CV9103 (CureVac), which encodes the PSA, PSCA, PSMA, and STEAP1 antigens for the treatment of patients with advanced prostate cancer [75]. In a Phase I/IIa clinical study involving 44 patients with advanced castration-resistant prostate cancer (CRPC), CV9103 was well tolerated and immunogenic, leading to an enhanced version of the vaccine, CV9104, which includes prostatic acid phosphatase (PAP) and mucin 1 (MUC1). The RNActive^®^ prostate cancer vaccine CV9104 was evaluated in a randomized, placebo-controlled Phase II trial to determine its clinical efficacy in prolonging survival for patients with asymptomatic or minimally symptomatic castrate-resistant metastatic prostate cancer [76]. However, the trial was terminated because the results showed no recent updates indicating a significant difference in overall survival (OS) between the treatment and control groups [77]. For a comprehensive list of ongoing clinical trial data involving mRNA vaccines for cancers (see Appendix A).

### 4.3. Personalized Medicine

The Human Genome Project has set the foundation for personalized medicine by providing a comprehensive mapping of the human genome. This mapping illustrates that an individual possesses a unique genetic makeup, which is valuable information for disease prevention, prognosis, diagnosis, and treatment [78]. Personalized medicine, also known as “precision medicine”, has revolutionized patient treatment by overcoming the limitations of conventional pharmacotherapies [79]. In personalized medicine, individualized treatment can be designed based on the specific characteristics of the individual identified through genomic investigations, specifically pharmacogenetic therapies. Lately, cancer immunotherapy has harnessed the specific characteristics of each tumour, developing vaccines and treatments with personalized approaches. mRNA vaccines have become a groundbreaking technology in oncology and personalized medicine, showing great potential for treating cancer. Their quick production and adaptability significantly advance immunotherapy [80]. mRNA vaccines targeting either key tumour-associated antigens (TAA) or tumour-specific antigens (TSA) have been developed [77,81] and have shown promising results. Tumour-specific antigens (TSA) or neoantigens provide a robust vaccination, providing better tumour specificity, and are not known to generate autoimmunity [82]. Because of their uniqueness and specificity to cancer cells, rationally, ongoing clinical trial outcomes have favoured more TSA vaccine development [69,83]. Most cancer mRNA personalized vaccines are TSA-based. TSA-based vaccines combine large-scale sequencing, bioinformatics algorithms, and patient immunology.

Recently, Merck and Moderna developed an individualized neoantigen therapy (INT) approach for treating patients with resected high-risk melanoma with mRNA-4157 (V940) in combination with pembrolizumab—an antibody that helps immune cells to oescape cancer cell suppression by binding to PD-1 on T cells [84]. This approach demonstrated clinically meaningful improvements in both recurrence-free survival and distant metastasis-free survival (62% reduction in the risks of distant metastasis and death) compared to pembrolizumab monotherapy, a contrast to the traditional checkpoint inhibitors with evidence of melanoma recurrence [84,85]. The mRNA-4157 (V940) is in a Phase III clinical trial stage. BioNTech recently published three-year follow-up data from a Phase 1 trial, which showed that 8 out of 16 patients exhibited sustained polyspecific T cell responses in patients with resected pancreatic ductal adenocarcinoma. This was associated with a delayed recurrence of tumours over three years following treatment with their mRNA (iNeST) or BNT122 vaccine, developed in collaboration with Genentech [86], encouraging the initiation of a Phase II trial [81].

Furthermore, Gritstone Bio’s neoantigen cancer vaccine GRANITE (GRT-C901/GRT-R902) Phase 2/3 study indicates a potential for clinical benefits for colorectal cancers in combination with immune checkpoint inhibitors [87]. Other personalized mRNA vaccines, such as Frame Therapeutic FRAME-001, are in a Phase II trial for the treatment of III-IV non-small cell lung cancer [88]. Indeed, the emergence of mRNA vaccine technology represents a significant shift in personalized cancer immunotherapy. Continual advances in high-throughput sequencing, neoantigen prediction, and nanocarrier systems sustain this strategy. Also, combining these vaccines with immune checkpoint inhibitors enhances their efficacy against tumour immune evasion. As these technologies improve, pharmacogenomics is expected to become cost-effective and more practical, promoting wider adoption of precision immuno-oncology in the future.

## 5. Current Challenges

### 5.1. Manufacturing and Scale-Up

#### 5.1.1. Challenges in Production and Supply Chain Scalability

Over the last decade and a half, mRNA vaccine development has undergone significant progress. The emergence of the Coronavirus disease 2019 (COVID-19) pandemic put the mRNA technology for vaccines to the test, and this was a success story with the first production of mRNA vaccines in less than a year of the pandemic [89]. This has resulted in a significant reduction in the time of production, cost, and ease of vaccine production, leading to a more rapid response to disease outbreaks, as demonstrated during the COVID-19 pandemic. There have been efforts to improve the technology further as demand for mRNA vaccines continues to increase. Wei et al., worked on the development of a universal integrated platform aimed at streamlining and enhancing on-demand preparation [90]. More advances in artificial intelligence have been propelling mRNA technology further; however, some challenges may persist, especially with differential global capacity.

#### 5.1.2. Addressing Regional Manufacturing Gaps

Vaccine production capacity is limited in most LMICs; for example, there are 11 vaccine manufacturing facilities in Africa, but their production capacities could hardly meet up to 1% vaccines for the over 50 countries in the continent. There are significant deficiencies in infrastructure, technology, and human capital [91]. Conscious efforts and deliberate investment will be required to reverse this, and the African Union (AU) should lead the continental efforts.

### 5.2. Cost and Accessibility

Generally, the production of mRNA vaccines has been documented to be cheaper when compared with traditional vaccine production. mRNA vaccines possess the advantage of fast manufacturing, as short as half a year, in contrast to the traditional method, which may take 10–15 years. Notwithstanding these foregoing merits, the mRNA vaccine requires higher capital and starting materials. This becomes more significant in the low- and middle-income countries that do not have adequate resources (material and human) to compete favourably with countries that are well-developed [92]. This challenge may further worsen the vaccine inequality that characterized COVID-19 vaccination at the peak of the pandemic between the global north and south. For example, Africa only supplies 1% of global vaccines [93]. The implication of this is that Africa, which constitutes almost 20% of the world, will have to continue to depend on other nations to meet its vaccine needs.

#### 5.2.1. High Costs of mRNA Technology Compared to Traditional Vaccines

There should be concerted efforts to ensure global vaccine equity. This could be led by the World Health Organization and other vaccine multinational groups like GAVI, the Vaccine Alliance. This may include, but is not limited to, technology transfer and licence sharing. Health system strengthening and improvement is another important factor in vaccine equity, especially in many low- and middle-income countries [94].

#### 5.2.2. Strategies for Reducing Costs and Ensuring Global Equity

Global equity for vaccines is not just right but is in the best interest of global health. It is important to learn from the lessons of COVID-19 vaccine nationalism and other barriers to the equitable distribution of mRNA vaccines, especially in LMIC. For example, local manufacturing of vaccines in Africa requires urgent attention and investment. Africa consumes one-quarter of the global vaccines, but 99% of these vaccines are produced outside of the continent. Scaling up local capacities for vaccine manufacturing will reverse this. The steps to take will include technology transfer, regulatory system strengthening, training, and workforce development [91]. In the long term, these efforts will drastically reduce the cost of vaccines and ensure global equity.

### 5.3. Safety Concerns and Public Acceptance

#### 5.3.1. Addressing Adverse Effects (e.g., Myocarditis)

The novelty of SARS-CoV-2 at the start of COVID-19 led to many uncertainties, not just about the virus but also about the introduction of the COVID-19 vaccine, which was developed at an unprecedented speed of less than a year. The implementation and emergency authorization of the new technology for the utilization of mRNA vaccines in humans for the first time raised many concerns. One such concern was the adverse events that could be associated with the new vaccines. While overall the mRNA vaccines for COVID-19 have been shown to be safe, there was a report of unusual adverse events. In a multinational study of over 99 million participants with combined doses of approximately 220 million of two COVID-19 mRNA vaccines (BNT162b2 and mRNA-1273), there was confirmation of pre-established safety signals relating to the cardiovascular system (myocarditis and pericarditis), neurological system (acute disseminated encephalomyelitis), and the identification of more safety signals including facial Bell’s palsy, febrile and generalized seizure, idiopathic thrombocytopenia, cerebral venous sinus thrombosis, splanchnic vein thrombosis, and pulmonary embolism [95]. It is important to further assess these findings in association studies rather than observational cohorts and to determine the clinical significance and implications.

In a small case–control study reported by [96], post-COVID-19 mRNA vaccine myocarditis was linked with significant plasma free full-length spike protein. The long-term implications of this will require further evaluation [96]. Descriptive terms for the acute and chronic adverse effects related to the COVID-19 mRNA vaccine as acute and post-acute COVID-19 vaccination syndrome. These terms are becoming well established [97]. Adverse events that are specific for lipid nanoparticles also warrant attention. Kent and his colleagues reported post-vaccination peak of mRNA and its ionizable lipid within 2 days with detection up to 4 weeks in some instances. The recirculation of mRNA lipid nanoparticles in the blood and their subsequent clearance by phagocytic cells may influence the poly-ethylene glycol (PEG) immunogenicity of the vaccine [98]. For a comprehensive review of these concerns, we refer readers to Bitounis et al. Addressing this concerns have been reviewed by Bitounis et al. [99].

In another trial setback, the FDA has put on hold all RSV vaccine trials in small children, citing an increase in severe illness following the negative outcome of a clinical trial conducted by Moderna among children aged 2–5 years old [100], Furthermore, two clinical trials on DC-based mRNA vaccines for cancers were terminated by Oslo University Hospital. However, no information has been provided for these terminations.

#### 5.3.2. Public Scepticism in the Wake of Misinformation

The global mobilization of human and material resources to tackle the COVID-19 pandemic faced the challenge of misinformation and disinformation, resulting in public scepticism, vaccine hesitancy, and anti-vaccine ideologies. Some of the related scepticism includes mutation or genetic changes posed by the vaccines, induction of autism, the vaccines were chips, and that they could cause injury and death [101]. Social media were instrumental to the spread of misinformation and disinformation on the vaccines. The WHO and other global and local health authorities, however, rose to the occasion to address this challenge.

### 5.4. Regulatory and Ethical Considerations

#### 5.4.1. Regulatory Landscape for mRNA Vaccines Beyond COVID

The application of mRNA vaccines has expanded beyond their use in infectious diseases, there are various mRNA technology uses in immunotherapy and cancer. Regulatory oversight will be very critical to ensure consistent ethical application of the rapidly evolving technology. There are no formal regulatory guidelines that are specific to mRNA vaccines, according to the World Health Organization, although key regulations for manufacturing and quality control are in place. Similarly, no regulatory guideline exists in the European Union, while the United Kingdom and United States regulation is based on the Human Medicines Regulation of 2012 and the Food and Drug Administration (FDA), respectively [102]. These are the same as regulatory bodies in other parts of the world. This is an important gap that needs to be addressed and tackled on time.

#### 5.4.2. Ethical Concerns Related to Personalized mRNA Vaccines

The application of personalized mRNA cancer vaccines is becoming a reality, and the issue of associated issues such as cost and equitable access, data privacy, and potential unintended side effects, especially in the long term. The concern of vaccine nationalism, as it was with the COVID-19 vaccine, may also resurface here [103]. The issue of technology transfer to LMIC is given the enormous resources that are associated with the development of these vaccines. All these and other related matters will require attention.

## 6. Innovations in mRNA Vaccine Designs

The technology of mRNA therapeutics has been in existence for the past few decades, but not until the COVID-19 pandemic that its fame spread after Pfizer and Moderna announced their mRNA vaccine for COVID-19. Stemirna Therapeutics is an mRNA-based technology company, established in 2016 to maximize the potential of RNA science. In spite of the successes recorded with the use of LNP for mRNA vaccine delivery during COVID-19, the technology faces many limitations. They are sensitive to temperature fluctuations and require ultra-cold storage (between −20 and to −80 °C). Some LNP components can trigger innate immune responses, leading to inflammation and adverse reactions. For a detailed review on current strategies to reduce the risks of mRNA vaccine toxicity, we recommend [104]. Following the development and successful delivery of the mRNA COVID-19 vaccine using lipid nanoparticles (LNP), there have been several attempts to improve and design new platforms that offer efficient and more targeted delivery of vaccines with reduced immune toxicity. Some of these strategies are summarised in Figure 2 below.

### 6.1. Multivalent Vaccine Designs for Multiple Pathogens or Cancer Antigens

Viruses are known for their unique ability to mutate rapidly. This variability within the same species of viruses reduces the ability of monovalent vaccines to protect against different serotypes of some viruses of global significance. The emergence of variants of concern (VOCs) necessitates the proposal for booster doses for the COVID-19 pandemic. Another example is the influenza virus. One of the major challenges for the development of a vaccine for the influenza virus is the multiple subtypes with varying degrees of mismatches. Following recent advances in nucleic acid-based vaccines, however, Arevalo and colleagues have developed a multivalent vaccine that encodes all 20 known subtypes [104]. This method also has applications in cancer immunotherapy.

Currently, numerous ongoing clinical trials are investigating the potential and immunogenic properties of several mRNA vaccines encoding tumour-specific neoantigen (see Table 2) for specific cancers. Personalized neoantigen vaccines (for example, BioNTech’s BNT111) target patient-specific tumour mutations. A 2021 trial by Sahin et al. reported prolonged survival in melanoma patients [105].

### 6.2. Future Direction

The current trend of designing multivalent mRNA vaccines for infectious diseases is a laudable approach that will help to protect millions of people across the world. Considering the number of ongoing clinical trials involving the use of dendritic cell-based mRNA vaccines, especially against cancers, one other approach that will be worth exploring is to load tumour-specific neoantigen mRNA on multiple combined dendritic cells (for example, pDC and mDC). This will most likely increase immunological response by activating more MHC molecules, and invariably, activation of cytotoxic T cells to target cancer cells (see Figure 3). Beside activating MHC I and II, DCs can provide signals for co-stimulatory molecules and cytokine secretion, ensuring specific T cell activation [106]. Moreover, these DCs are considered to be relatively easy to propagate ex vivo. One important tool that will be valuable in the design of more targeted and personalized mRNA vaccines is the use of AI.

## 7. Conclusions

It is encouraging to see the level of progress that has been made over the past few years since the emergence of the mRNA vaccine. Despite public scepticism, the available clinical trial data reveal a profound justification for continuous research and collaboration to address global health challenges. With an increasing number of clinical trials in the use of mRNA vaccines, particularly not just against infectious diseases but also cancer, this technology will usher in a new era in vaccinology. The collaboration between companies to optimize different structural components of the vaccine should be replicated among universities and institutions’ mRNA experts in order to overcome current challenges facing this area of research. Finally, it would be desirable to see more funding support from the government and all stakeholders to ensure equity and global distribution of this technology to people who need it the most.

## Figures and Tables

**Figure 1 vaccines-13-00601-f001:**
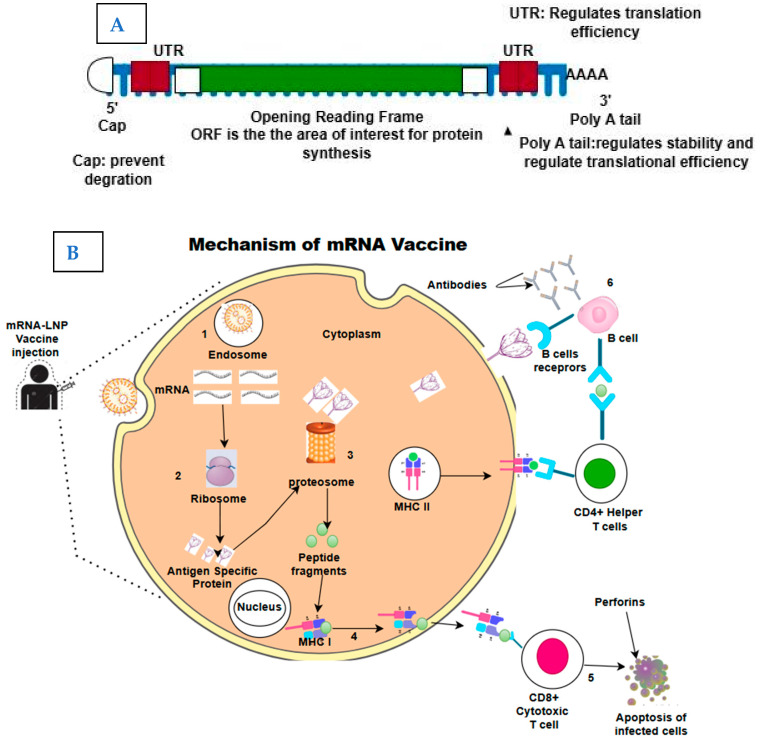
(**A**), an illustration of mRNA structure. (**B**): The mechanism of mRNA vaccine in vivo. 1. After injection, the LNP-encapsulated vaccine is internalized by APCs like dendritic cells and macrophages through endocytosis. 2. In the cytoplasm, the mRNA is released to be translated into a tumour-specific antigen in the ribosome. 3. The synthesized antigens are processed into peptide fragments by the proteosome. 4. The peptide fragments are loaded on major histocompatibility complex (MHC) molecules. MHC class I molecules present the peptides to CD8+ cytotoxic T cells, while MHC class II molecules present them to CD4+ helper T cells. 5. The interactions between MHC-antigen complexes and T cell receptors (TCRs) activate T cells, which target and destroy tumour cells expressing the same antigen. The MHC II-antigen complex activates helper T cells, which in turn activate B cells. 6. The interaction between helper T cells activates the B cell for antibody production, and/or the B cell recognizes tumour-specific antigen on the cell surface and forms a complex via B cell receptors (BCRs) to activate B cells and lead to the production of antibodies.

**Figure 2 vaccines-13-00601-f002:**
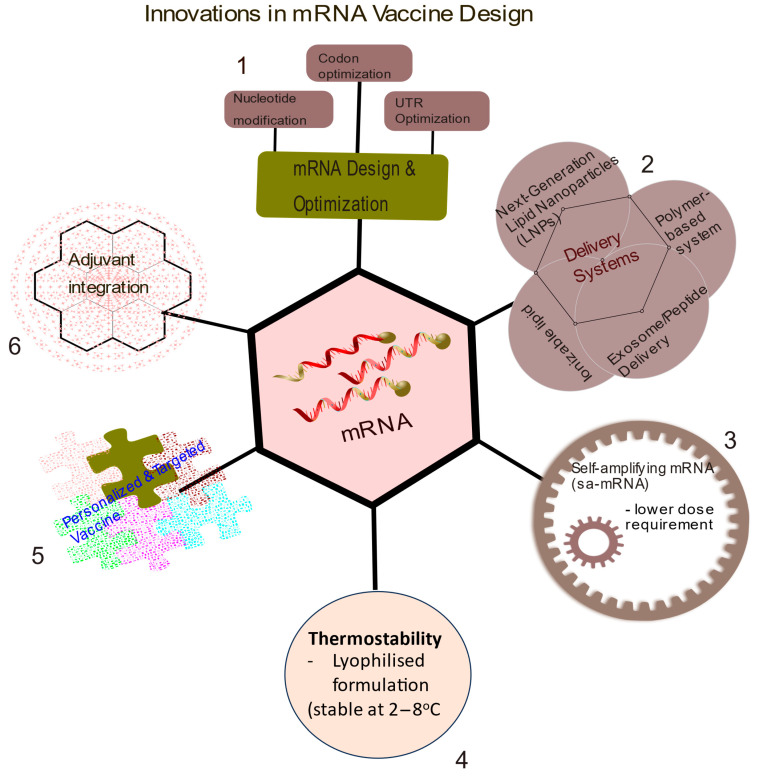
Summary of current innovations in mRNA vaccine design. 1. Structural optimization of mRNA to enhance packaging and delivery. 2. New methods aimed at improving delivery systems over the traditional lipid nanoparticles, such as the use of exosome/peptide, ionizable lipids for easy processing, and the release of mRNA molecules upon injection. 3. Self-amplifying mRNA (sa-mRNA) for lower dose requirement of mRNA vaccine, as the incorporated molecules like zapomeran help to produce more copies of mRNA. 4. Thermostability is another strategy to overcome the need for low temperature for the storage and transportation of mRNA-LNPs vaccines. These vaccines are stable at temperatures of about 2–8 °C. 5. Personalized and targeted vaccines based on patients’ genetic profiles. 6. Adjuvant integration of additional molecules like cytokines or a combination of vaccines with traditional immune blockade therapies.

**Figure 3 vaccines-13-00601-f003:**
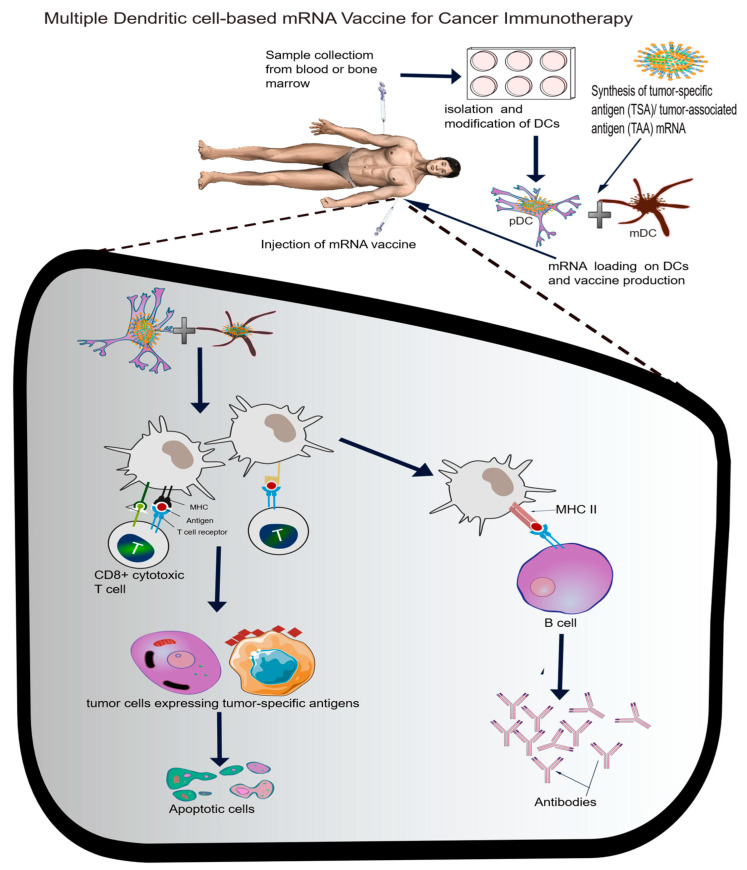
Scheme for multiple dendritic cell-based mRNA vaccine against cancers. The combination of myeloid DC (mDC) and plasmacytoid DC (pDC) could significantly improve immune activation and antibody production against cancer.

**Table 1 vaccines-13-00601-t001:** Ongoing clinical trials on infectious diseases.

Company	Candidate	Product Type	Disease	Status	CT Ref. (Accessed on 26 May 2025)
Sanofi, France	VAV00027	mRNA/LNP	RSV/hMPV	Phase I	NCT06237296 (https://clinicaltrials.gov/study/NCT06237296?cond=Infectious%20Diseases&intr=mRNA%20Vaccine&aggFilters=status:not%20rec%20act%20ter%20com&rank=2)
	VBS00001	mRNA vaccine	Pandemic flu H5	Phase I/II	NCT06727058 (https://clinicaltrials.gov/study/NCT06727058?cond=Infectious%20Diseases&intr=mRNA%20Vaccine&aggFilters=status:not%20rec%20act%20ter%20com&rank=10)
	MRT5421, MRT5424, and MRT5429	mRNA vaccine	Influenza	Phase I/II	NCT06361875 (https://clinicaltrials.gov/study/NCT06361875?intr=mRNA%20Vaccine&page=2&rank=17)
		mRNA vaccine	Influenza	Phase I/II	NCT06744205 (https://clinicaltrials.gov/study/NCT06744205?intr=mRNA%20Vaccine&page=2&rank=18)
	mRNA-NA	mRNA vaccine	Influenza	Phase I	NCT05426174 (https://clinicaltrials.gov/study/NCT05426174?intr=mRNA%20Vaccine&page=3&rank=24)
		mRNA vaccine	Influenza	Phase I	NCT05829356 (https://clinicaltrials.gov/study/NCT05829356?intr=mRNA%20Vaccine&page=3&rank=27)
		mRNA vaccine	Acne	Phase I/II	NCT06316297 (https://clinicaltrials.gov/study/NCT06316297?intr=mRNA%20Vaccine&page=4&rank=32)
	Rabipur^®^ CV7202	mRNA vaccine	Rabies	Phase I	NCT03713086 (https://clinicaltrials.gov/study/NCT03713086?intr=mRNA%20Vaccine&page=4&rank=35)
	RNActive^®^ CV7201	mRNA vaccine	Rabies	Phase I	NCT02241135 (https://clinicaltrials.gov/study/NCT02241135?intr=mRNA%20Vaccine&page=5&rank=47)
	MRT5413	mRNA vaccine	Influenza	Phase I/II	NCT05650554 (https://clinicaltrials.gov/study/NCT05650554?intr=mRNA%20Vaccine&page=7&rank=61)
		mRNA vaccine	Influenza	Phase I	NCT06118151 (https://clinicaltrials.gov/study/NCT06118151?intr=mRNA%20Vaccine&page=6&rank=59)
Moderna TX, Inc., USA	mRNA-1189	mRNA vaccine	Epstein–Barr Virus (EBV)	Phase I/II	NCT05164094 (https://clinicaltrials.gov/study/NCT05164094?rank=1)
	mRNA-1644 mRNA-1574		HIV	Phase I/II	NCT03547245 (https://www.clinicaltrials.gov/study/NCT03547245)
	mRNA-1215		Nipah Virus (NiV)	Phase I	NCT05398796 (https://clinicaltrials.gov/study/NCT05398796)
	mRNA-1769		Mpox	Phase I/II	NCT05995275 (https://clinicaltrials.gov/study/NCT05995275?rank=1)
	mRNA-1403		Novovirus/(Acute Gastroenteritis)		NCT06592794 (https://clinicaltrials.gov/study/NCT06592794?rank=1)
	mRNA-1647		Cytomegalovirus (CMV)	Phase III	NCT05085366 (https://clinicaltrials.gov/study/NCT05085366?cond=Infectious%20Diseases&intr=mRNA%20Vaccine&aggFilters=status:not%20rec%20act%20ter%20com&page=8&rank=72)
	mRNA-1325, mRNA-1893		Zika virus	Phase I Phase II	NCT03014089 (https://clinicaltrials.gov/study/NCT03014089?cond=Infectious%20Diseases&intr=mRNA%20Vaccine&aggFilters=status:not%20rec%20act%20ter%20com&page=7&rank=63) NCT04064905 (https://clinicaltrials.gov/study/NCT04064905?rank=1)
	mRNA-1010		Seasonal Flu	Phase III	NCT06602024 (https://clinicaltrials.gov/study/NCT06602024?rank=1)
	mRNA-1011.1, mRNA-1011.2, mRNA-1012.1	Next-Gen mRNA vaccines	Influenza Virua	Phase I/II	NCT05827068 (https://clinicaltrials.gov/study/NCT05827068?rank=1)
	mRNA-1345	mRNA vaccine	Respiratory Syncytial Virus (RSV)	Phase III	NCT06067230 (https://clinicaltrials.gov/study/NCT06067230?rank=1)
	mRNA-1365		Human Metapneumovirus (hMPV)	Phase I	NCT05743881 (https://clinicaltrials.gov/study/NCT05743881?rank=1)
	mRNA-1608		Herpes Simplex Virus	Phase I/II	NCT06033261 (https://clinicaltrials.gov/study/NCT06033261?rank=1)
	mRNA-1468		Herpes Zoster (HZ)	Phase I/II	NCT05701800 (https://clinicaltrials.gov/study/NCT05701800?rank=1)
	VAL-339851		Seasonal influenza	Phase I	NCT03345043 (https://clinicaltrials.gov/study/NCT03345043?rank=1)
	mRNA-1944		Chikungunya Virus	Phase I	NCT03829384 (https://clinicaltrials.gov/study/NCT03829384?rank=1)
CanSino Biologics, China		mRNA vaccine	COVID-19	Phase I/II	NCT05373485 (https://clinicaltrials.gov/study/NCT05373485?intr=mRNA%20Vaccine&page=4&rank=33) NCT05373472 (https://clinicaltrials.gov/study/NCT05373472?intr=mRNA%20Vaccine&page=4&rank=34)
National Institute of Allergy and Infectious Diseases (NIAID), USA	HVTN 302	BG505 MD39.3 mRNA/BG505 MD39.3 gp151 mRNA/BG505 MD39.3 gp151 CD4KO mRNA	HIV	Phase I	NCT05217641 (https://clinicaltrials.gov/study/NCT05217641?cond=Infectious%20Diseases&intr=mRNA%20Vaccine&aggFilters=status:not%20rec%20act%20ter%20com&rank=4)
	mRNA-1273	mRNA-LNP	COVID-19	Phase I	NCT04283461 (https://clinicaltrials.gov/study/NCT04283461?cond=Infectious%20Diseases&intr=mRNA%20Vaccine&aggFilters=status:not%20rec%20act%20ter%20com&page=7&rank=69)
	mRNA-1215	mRNA vaccine	Nipah Virus (NiV)	Phase I	NCT05398796 (https://clinicaltrials.gov/study/NCT05398796?cond=Infectious%20Diseases&intr=mRNA%20Vaccine&aggFilters=status:not%20rec%20act%20ter%20com&page=2&rank=14)
Massachusetts General Hospital, USA		mRNA-transfected autologous dendritic cells	HIV-1	Phase I/II	NCT00833781 (https://clinicaltrials.gov/study/NCT00833781?cond=Infectious%20Diseases&intr=mRNA%20Vaccine&aggFilters=status:not%20rec%20act%20ter%20com&rank=7)
AstraZeneca, UK	AZD9838/AZD6563	mRNA-VLP	SARS-CoV-2	Phase I	NCT06147063 (https://clinicaltrials.gov/study/NCT06147063?cond=Infectious%20Diseases&intr=mRNA%20Vaccine&aggFilters=status:not%20rec%20act%20ter%20com&rank=9)
CSPC ZhongQi Pharmaceutical Technology Co., Ltd., China	SYS6006	mRNA vaccine	SARS-CoV-2	Phase II	NCT05439824 (https://clinicaltrials.gov/study/NCT05439824?cond=Infectious%20Diseases&intr=mRNA%20Vaccine&aggFilters=status:not%20rec%20act%20ter%20com&page=3&rank=21)
Immorna Biotherapeutics, Inc., China	JCXH-221	mRNA vaccine	COVID-19	Phase II	NCT05743335 (https://clinicaltrials.gov/study/NCT05743335?cond=Infectious%20Diseases&intr=mRNA%20Vaccine&aggFilters=status:not%20rec%20act%20ter%20com&page=3&rank=28)
	JCXH-108	mRNA vaccine	RSV	Phase I	NCT06564194 (https://clinicaltrials.gov/study/NCT06564194?cond=Infectious%20Diseases&intr=mRNA%20Vaccine&aggFilters=status:not%20rec%20act%20ter%20com&page=4&rank=31)
CNBG-Virogin Biotech (Shanghai) Ltd., China	ZSVG-02-O	mRNA vaccine	COVID-19	Phase II	NCT06113731 (https://clinicaltrials.gov/study/NCT06113731?cond=Infectious%20Diseases&intr=mRNA%20Vaccine&aggFilters=status:not%20rec%20act%20ter%20com&page=3&rank=29)
Vaxart Biosciences, USA	VXA-CoV2-3.1	Oral SARS-CoV-2 Vaccine Tablet	COVID-19	Phase II	NCT06672055 (https://clinicaltrials.gov/study/NCT06672055?cond=Infectious%20Diseases&intr=mRNA%20Vaccine&aggFilters=status:not%20rec%20act%20ter%20com&page=4&rank=33)
Shenzhen Shenxin Biotechnology Co., Ltd., China	IN001	mRNA vaccine	Herpes Zoster	Phase I	NCT06375512 (https://clinicaltrials.gov/study/NCT06375512?cond=Infectious%20Diseases&intr=mRNA%20Vaccine&aggFilters=status:not%20rec%20act%20ter%20com&page=4&rank=40)
Lemonex Inc., South Korea	LEM-mR203	mRNA-DegradaBALL vaccine	COVID-19	Phase I	NCT06032000 (https://clinicaltrials.gov/study/NCT06032000?cond=Infectious%20Diseases&intr=mRNA%20Vaccine&aggFilters=status:not%20rec%20act%20ter%20com&page=5&rank=41)
Barinthus Biotherapeutics, UK	ChAdOx1-HBV	mRNA HBV vaccine	Chronic HBV infection	Phase I	NCT04297917 (https://clinicaltrials.gov/study/NCT04297917?cond=Infectious%20Diseases&intr=mRNA%20Vaccine&aggFilters=status:not%20rec%20act%20ter%20com&page=5&rank=43)
RinuaGene Biotechnology Co., Ltd., China	RG002	mRNA vaccine	HPV16/18 associated Cervical Intraepithelial Neoplasia Grade 2 or 3(CIN2/3).	Phase I/II	NCT06273553 (https://clinicaltrials.gov/study/NCT06273553?cond=Infectious%20Diseases&intr=mRNA%20Vaccine&aggFilters=status:not%20rec%20act%20ter%20com&page=5&rank=46)
Speransa Therapeutics, USA		PRIME-2-CoV_Beta	COVID-19	Phase I	NCT05367843 (https://clinicaltrials.gov/study/NCT05367843?cond=Infectious%20Diseases&intr=mRNA%20Vaccine&aggFilters=status:not%20rec%20act%20ter%20com&page=6&rank=52)
Albert B. Sabin Vaccine Institute, USA	Sinovac	mRNA vaccine	COVID-19	Phase IV	NCT05343871 (https://clinicaltrials.gov/study/NCT05343871?cond=Infectious%20Diseases&intr=mRNA%20Vaccine&aggFilters=status:not%20rec%20act%20ter%20com&page=6&rank=60)
Sinocelltech Ltd., China	SCTV01E-2	mRNA vaccine	COVID-19	Phase II	NCT05933512 (https://clinicaltrials.gov/study/NCT05933512?cond=Infectious%20Diseases&intr=mRNA%20Vaccine&aggFilters=status:not%20rec%20act%20ter%20com&page=6&rank=58)
SK Bioscience Co., Ltd., South Korea	GBP560-A GBP560-B	mRNA vaccine	SK Japanese Encephalitis virus disease	Phase I/II	NCT06680128 (https://clinicaltrials.gov/study/NCT06680128?cond=Infectious%20Diseases&intr=mRNA%20Vaccine&aggFilters=status:not%20rec%20act%20ter%20com&page=7&rank=61)
Gritstone bio, Inc., USA Seqirus, UK Stemirna Therapeutics, China	GRT-R912, GRT-R914, and GRT-R918	samRNA Vaccine	COVID-19/HIV	Phase I	NCT05435027 (https://clinicaltrials.gov/study/NCT05435027?cond=Infectious%20Diseases&intr=mRNA%20Vaccine&aggFilters=status:not%20rec%20act%20ter%20com&page=7&rank=68)
	V202_01	Sa-mRNA vaccine	Influenza	Phase I	NCT06028347 (https://clinicaltrials.gov/study/NCT06028347?intr=mRNA%20Vaccine&page=1&rank=3)
	SWC002	mRNA vaccine	COVID-19	Phase I/II	NCT05144139 (https://clinicaltrials.gov/study/NCT05144139?intr=mRNA%20Vaccine&page=1&rank=5)
	SW-BIC-213	mRNA vaccine	COVID-19	Phase III	NCT05580159 (https://clinicaltrials.gov/study/NCT05580159?intr=mRNA%20Vaccine&page=7&rank=65)
	SWIM816	mRNA vaccine	COVID-19	Phase II/III	NCT05911087 (https://clinicaltrials.gov/study/NCT05911087?intr=mRNA%20Vaccine&page=10&rank=99)

**Table 2 vaccines-13-00601-t002:** Multivalent vaccines in trials.

Company/Sponsor	Candidate	Product Type	Disease(s)	Status	Reference (Accessed on 26 May 2025)
BioNTech, Germany/Roche, USA	BNT112 (autogene cevumeran) vaccine, neoantigen	mRNA vaccine	Malignant melanoma	Phase II	NCT03815058 (https://clinicaltrials.gov/study/NCT03815058)
			Colorectal cancer	Phase II	NCT04486378 (https://clinicaltrials.gov/study/NCT04486378)
			Other metastatic tumours	Phase II	NCT03289962 (https://clinicaltrials.gov/study/NCT03289962)
			Pancreatic ductal adenocarcinoma (PDAC)	Phase II	NCT05968326 (https://clinicaltrials.gov/study/NCT05968326)
			Muscle invasive urothelial carcinoma (MIUC)	Phase II	NCT06534983 (https://clinicaltrials.gov/study/NCT06534983)
Moderna/Merck and Co., USA.	mRNA-4157 vaccine/neoantigen	mRNA vaccine	Bladder cancer	Phase I/II	NCT06305767 (https://clinicaltrials.gov/study/NCT06305767)
			non-small cell lung cancer (NSCLC)	Phase III	NCT06077760 (https://clinicaltrials.gov/study/NCT06077760)
			Renal cell carcinoma	Phase II	NCT06307431 (https://clinicaltrials.gov/study/NCT06307431)
			Malignant melanoma	Phase III	NCT05933577 (https://clinicaltrials.gov/study/NCT05933577)
			Others	Phase II/III	NCT06295809 (https://clinicaltrials.gov/study/NCT06295809)
	mRNA-1195	mRNA vaccine	Multiple Sclerosis	Phase II	NCT06735248 (https://clinicaltrials.gov/study/NCT06735248?rank=1)
	mRNA-3927		Propionic Acidemia	Phase I/II	NCT04159103 (https://clinicaltrials.gov/study/NCT04159103)
	mRNA-3745		Glycogen Storage Disease Type 1a (GSD1a)	Phase I/II	NCT05095727 (https://clinicaltrials.gov/study/NCT05095727?rank=1)
	mRNA-3705		methylmalonic acidemia (MMA)	Phase I/II	NCT05295433 (https://clinicaltrials.gov/study/NCT05295433?rank=1)
	mRNA-1083		COVID-19 + Influenza	Phase III	NCT06694389 (https://clinicaltrials.gov/study/NCT06694389?rank=1)
	mRNA-1975, mRNA-1982		Lyme Disease	Phase I/II	NCT05975099 (https://clinicaltrials.gov/study/NCT05975099?rank=1)
	mRNA-1653		hMPV + Parainfluenza Virus Type 3 (PIV3)	Phase I	NCT04144348 (https://clinicaltrials.gov/study/NCT04144348?rank=1)
	mRNA-1045		Influenza + RSV	Phase I	NCT05585632 (https://clinicaltrials.gov/study/NCT05585632?rank=1)
	mRNA-1230		Influenza + RSV + SARS-CoV-2	Phase I	NCT05585632 (https://clinicaltrials.gov/study/NCT05585632?rank=1)

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
