# Peer review of "mRNA Vaccine Technology Beyond COVID-19"

_vaccines, 2025, doi:10.3390/vaccines13060601_

Round 1

Reviewer 1 Report

Comments and Suggestions for Authors

This is a very interesting review article on mRNA vaccine technology beyond COVID-19. The review is well structured and written. The included table are very useful and the figures appropriate. The is no criticism from my side. As a suggestion, the authors may include also a paragraph on alternative delivery systems beyond the known lipid particles, i.e. polymers, polycations

Author Response

Comment: This is a very interesting review article on mRNA vaccine technology beyond COVID-19. The review is well structured and written. The included table are very useful and the figures appropriate. The is no criticism from my side. As a suggestion, the authors may include also a paragraph on alternative delivery systems beyond the known lipid particles, i.e. polymers, polycations

Response: Thank you for pointing this out. We agree with this observation. Therefore we have included a paragraph in the manuscript for a brief mention about the use of polymers and the different polycations some researchers have investigated for delivery of mRNA vaccine. We could not go in-depth as the length of our manuscript does not provide us much space to add a lot of information again.

This addition is now on page 6, lines 232-243.

"Cationic polymers, such as polyethyleneimine (PEI) and poly(amidoamine) (PAMAM), have numerous amino acids that provide a high density of positive charges for mRNA complexation and endosomal escape, hence a high delivery efficiency. It acts as a protonation buffer, allowing protons, chlorides, and water to enter the nucleus [12,35]. This causes osmotic expansion and more endosomal membrane disruption. Despite the delivery efficiency, they have high cytotoxicity levels. Non cationic polymers like polyethylene glycol (PEG) and cyclodextrin (CD), which are biocompatible compounds are conjugated to the cationic polymers to lower their charge density while maintaining their ability to condense nucleic acids, thus lowering the toxicity level. Considering the low biodegradability and biocompatibility of the current polymers, other polymeric carriers such as polyesters and dendrimers are being developed to overcome this limitation [35,36]."

Reviewer 2 Report

Comments and Suggestions for Authors

 In the review mRNA Vaccine Technology Beyond COVID-19 it was presented  the current state of mRNA vaccine technology and its use against other diseases. 

The article is too long and some information about cancer is repeated, it needs to be restructured  and  it must improved by changing of the  chapter order. 

Chapter 4.1 becomes Chapter 3 and is called LPL delivery system
Chapter 3 becomes Chapter 4 (4.1 Infectious diseases, 4.2 Cancer immunology, 4.3 Personalised medicine - explain the role of pembrolizumab - line 288 )

Chapter 5 Current challenges should also include Chapter 6 Regulatory and ethical considerations 
Chapter 4 becomes Chapter 6 Innovation in mRNA Vaccine designs (lines 311-325) which will also include Chapter 4.4. Multivalent vaccine designs for multiple pathogens or cancer antigens
All tables will be added as supplementary material, figures remain in the text

Comments on the Quality of English Language

Author Response

Comments 1: The article is too long and some information about cancer is repeated, it needs to be restructured  and  it must improved by changing of the  chapter order. 

Response 1: Thank you for pointing out this. We had this concern earlier before the first submission for review and your observation has helped us to thoroughly review the manuscript and duplicated information has been removed in the process.

Comments 2: Chapter 4.1 becomes Chapter 3 and is called LPL delivery system
Chapter 3 becomes Chapter 4 (4.1 Infectious diseases, 4.2 Cancer immunology, 4.3 Personalised medicine - explain the role of pembrolizumab - line 288 )

Response 2: We agree. However, we hope the reviewer meant Lipid Nanoparticles (LNP) delivery system instead of LPL delivery system. Chapter 3 now contain information about Lipid Nanoparticles (lines 140-291). 

Now, Chapter 4 is Applications Beyond COVID-19 (lines 292-460). The information about the role of pembrolizumab is included in lines 438-439.

Comments 3: Chapter 5 Current challenges should also include Chapter 6 Regulatory and ethical considerations 
Chapter 4 becomes Chapter 6 Innovation in mRNA Vaccine designs (lines 311-325) which will also include Chapter 4.4. Multivalent vaccine designs for multiple pathogens or cancer antigens.

Response 3: Regulatory and ethical consideration has been added to Chapter 5 Current challenges (lines 567-589). Chapter 6 now contain information about Innovation in mRNA Vaccine Designs, and have Chapter 6.1 Multivalent vaccine designs for multiple pathogens or cancer antigens (lines 591-654).

Comments 4: All tables will be added as supplementary material, figures remain in the text

Response 4: While we sincerely appreciate this suggestion, we do not think it will be appropriate to have no table in the article. Hence, we have removed the longest table (List of ongoing clinical trials involving mRNA vaccine therapies in cancers) which was in line 510, and attached it as supplementary material. We believe this should be sufficient to keep the article at an average length. 

Reviewer 3 Report

Comments and Suggestions for Authors

The article is an interesting and comprehensive review of the various RNA vaccine trials in infectiology, oncology and immunopathology. It needs to be improved on a number of points:
- In chapter 2.1, the authors should really mention the addition of pseudouridine as this is the keystone of the pioneers' invention (they should cite the original reference ie Karikó K, Buckstein M, Ni H, Weissman D. Suppression of RNA recognition by Toll-like receptors: the impact of nucleoside modification and the evolutionary origin of RNA. Immunity. 2005 Aug;23(2):165-75. doi: 10.1016/j.immuni.2005.06.008. PMID: 16111635.)
- The authors have got to mention the potential unexpected effects of mRNA vaccines, particularly in the COVID experiment, notably the presence of the vaccine protein (Spike for the COVID vaccine) which persists in circulating blood and may be associated with myocarditis (see Yonker LM et al. Circulating Spike Protein Detected in Post-COVID-19 mRNA Vaccine Myocarditis. Circulation. 2023 doi: 10.1161/CIRCULATIONAHA.122.061025.

For a review see Scholkmann F, May CA.COVID-19, syndrome post-aiguë COVID-19 (PACS, « long COVID ») et syndrome post-COVID-19 (PCVS, « post-COVIDvac-syndrome ») : Similitudes et différences. Pathol Res Pract. 2023 Jun;246:154497. doi : 10.1016/j.prp.2023.154497.

- There are also other effects as shown by S. J., et al. (2024). Blood Distribution of SARS-CoV-2 Lipid Nanoparticle mRNA Vaccine in Humans. ACS Nano. doi.org/10.1021/acsnano.4c11652. This important study revealed how mRNA and its fatty nanoparticle shell peaked in the bloodstream within two days after vaccination, and  in some cases, the mRNA remains detectable for up to a month. This suggests that the circulation of mRNA lipid nanoparticles into the blood and their clearance by phagocytes influence the PEG immunogenicity of the mRNA vaccines, and consequently their safety.
- Lipid nanoparticle structural components are the key compounds of the mRNA vaccines: they all present toxicity concerns and lipid nanoparticles can lead to toxicity, and their possible reactogenicity (see Bitounis D, Jacquinet E, Rogers MA, Amiji MM. Strategies to reduce the risks of mRNA drug and vaccine toxicity. Nat Rev Drug Discov. 2024 Apr;23(4):281-300. doi: 10.1038/s41573-023-00859-3. Epub 2024 Jan 23. PMID: 38263456)

Consequently the authors must cite these studies in their article.

- The MS vaccine cited (NCT06735248) targets EBV

Comments on the Quality of English Language

English is correct

Author Response

Comments 1:  In chapter 2.1, the authors should really mention the addition of pseudouridine as this is the keystone of the pioneers' invention (they should cite the original reference ie Karikó K, Buckstein M, Ni H, Weissman D. Suppression of RNA recognition by Toll-like receptors: the impact of nucleoside modification and the evolutionary origin of RNA. Immunity. 2005 Aug;23(2):165-75. doi: 10.1016/j.immuni.2005.06.008. PMID: 16111635.)

Response 1: Thank you for pointing this out. We agree with this comment. Therefore, we have included the information about pseudouridine (lines 124-126), and cited the pioneers' work as [20] in the reference.

Comments 2: The authors have got to mention the potential unexpected effects of mRNA vaccines, particularly in the COVID experiment, notably the presence of the vaccine protein (Spike for the COVID vaccine) which persists in circulating blood and may be associated with myocarditis (see Yonker LM et al. Circulating Spike Protein Detected in Post-COVID-19 mRNA Vaccine Myocarditis. Circulation. 2023 doi: 10.1161/CIRCULATIONAHA.122.061025.

Response 2: We agree. Therefore, we have included information about the challenges of COVID vaccine as regards the potential presence of Spike protein in the blood as reported by Yonker et. al (lines 539-541), and Yonker et. al is cited as [97].

Comments 3: There are also other effects as shown by S. J., et al. (2024). Blood Distribution of SARS-CoV-2 Lipid Nanoparticle mRNA Vaccine in Humans. ACS Nano. doi.org/10.1021/acsnano.4c11652. This important study revealed how mRNA and its fatty nanoparticle shell peaked in the bloodstream within two days after vaccination, and  in some cases, the mRNA remains detectable for up to a month. This suggests that the circulation of mRNA lipid nanoparticles into the blood and their clearance by phagocytes influence the PEG immunogenicity of the mRNA vaccines, and consequently their safety.
Lipid nanoparticle structural components are the key compounds of the mRNA vaccines: they all present toxicity concerns and lipid nanoparticles can lead to toxicity, and their possible reactogenicity (see Bitounis D, Jacquinet E, Rogers MA, Amiji MM. Strategies to reduce the risks of mRNA drug and vaccine toxicity. Nat Rev Drug Discov. 2024 Apr;23(4):281-300. doi: 10.1038/s41573-023-00859-3. Epub 2024 Jan 23. PMID: 38263456)

Response 3: We agree. Therefore, all these effects have been included in a summarised manner as the length of the article could permit us and the authors cited appropriately (lines 544-550) and all references cited. Scholkmann et. al is cited [98] while Bitounis et. al is [100].

Comments 4: The MS vaccine cited (NCT06735248) targets EBV

Response 4: We do not agree with this comments. We encourage the reviewer to check the record of this clinical trial on ClinicalTrials.gov via this link: Study Details | A Study to Investigate Multiple Sclerosis Relapse Prevention With mRNA-1195 Compared With Placebo in Participants Aged 18 to ≤55 Years | ClinicalTrials.gov. Hence, we did not make any changes to this entry in the manuscript.